# Magnetic impurity in a one-dimensional few-fermion system

Lukas Rammelmüller[1,2*], David Huber[3], Matija Čufar[4], Joachim Brand[4], Hans-Werner Hammer[3,5], Artem G. Volosniev[6†]

**1** Arnold Sommerfeld Center for Theoretical Physics (ASC), University of Munich, Theresienstr. 37, 80333 München, Germany
**2** Munich Center for Quantum Science and Technology (MCQST), Schellingstr. 4, 80799 München, Germany
**3** Technische Universität Darmstadt, Department of Physics, 64289 Darmstadt, Germany
**4** Dodd-Walls Centre for Photonic and Quantum Technologies and Centre for Theoretical Chemistry and Physics, New Zealand Institute for Advanced Study, Massey University, Auckland 0632, New Zealand
**5** ExtreMe Matter Institute EMMI and Helmholtz Forschungsakademie Hessen für FAIR (HFHF), GSI Helmholtzzentrum für Schwerionenforschung GmbH, 64291 Darmstadt, Germany
**6** IST Austria (Institute of Science and Technology Austria), Am Campus 1, 3400 Klosterneuburg, Austria
\* lukas.rammelmueller@physik.uni-muenchen.de
† artem.volosniev@ist.ac.at

April 5, 2022

## Abstract

We present a numerical analysis of spin-$\frac{1}{2}$ fermions in a one-dimensional harmonic potential in the presence of a magnetic point-like impurity at the center of the trap. The model represents a few-body analogue of a magnetic impurity in the vicinity of an $s$-wave superconductor. Already for a few particles we find a ground-state level crossing between sectors with different fermion parities. We interpret this crossing as a few-body precursor of a quantum phase transition, which occurs when the impurity 'breaks' a Cooper pair. This picture is further corroborated by analyzing density-density correlations in momentum space. Finally, we discuss how the system may be realized with existing cold-atoms platforms.

# 1 Introduction

Quantum phase transitions (QPT's) are transitions between different phases of a many-body quantum system at zero temperature. In a QPT the ground state of the many-body system changes qualitatively due to quantum fluctuations as an external control parameter is varied. Such a control parameter can, for example, be an external magnetic field or, in theoretical studies, simply a coupling constant in the Hamiltonian. QPT's play an important role in quantum many-body systems. They are typically studied in the context of a macroscopic number of constituents and linked to the collective behavior of many particles [1]. However, the qualitative behavior of many mesoscopic systems with a modest number of particles, such as atomic nuclei [2] and few cold fermions [3, 4], can also be understood using tools developed to study QPT's. This implies the possibility to study the emergence of QPT's from few-body dynamics, which falls into a broad class of studies dedicated to the so-called "few-body precursors" of many-body phenomena, see, e.g., [5–7].

    The transition from few-body behavior to many-body behavior as a function of the particle number has been explored theoretically in a broad variety of one-dimensional fermionic systems using exact diagonalization [8], Monte Carlo methods [9–11], coupled cluster expansion [12], and perturbation theory [13], see Refs. [5, 6] for a review. For similar studies in higher spatial dimensions, see, e.g., Refs. [3, 14–16]. The interest in this "bottom-up" approach to many-body physics is driven in particular by the existing ultracold atomic set-ups whose exquisite

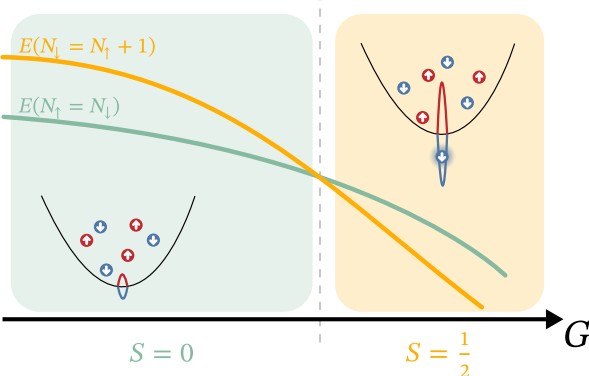

Figure 1: **Sketch of the transition driven by the magnetic impurity.** With increasing the strength of the impurity-fermion interaction, $G$, a level crossing between the sectors of different fermionic parity (i.e., odd vs. even particle number) occurs. At the same time, the magnetization of the ground state changes from $S = 0$ to $S = \frac{1}{2}$. The feature persists even in the presence of only a handful of particles and constitutes a precursor of a many-body QPT.

tunability and control admits the realization of experiments with a precisely determined small number of particles. Such setups have been used, for instance, to study the formation of a Fermi sea and a pairing gap in quasi one-dimensional few-fermion systems [17, 18]. Another example is given by the observation of a few-body counterpart of the Higgs amplitude mode across the normal to superfluid phase transition in a two-dimensional Fermi gas [4].

Motivated by these studies, we explore the few-body analogue of a classical magnetic impurity in an $s$-wave superconductor. In the many-body limit, this is a well-studied problem [19, 20], which has been found to host a sharp QPT from a non-magnetic total spin $S = 0$ ground state (assuming that the spin of the impurity is zero) for weakly interacting impurities to a $S = \frac{1}{2}$ ground state at strong impurity-electron interactions [21], see a sketch in Fig. 1. This transition, which manifests itself as a crossing of the energy levels corresponding to the two ground states, is connected to the so-called Yu-Shiba-Rusinov states (or simply Shiba states) – excited states below the threshold of the single-particle gap (so-called sub-gap states) [22–24].

In this work, we focus on the emergence of this QPT in a one-dimensional (1D) few-body system that can be simulated using cold-atom set-ups. Specifically, we investigate the few-body sector of a two-component Fermi gas in a 1D harmonic trap. The static magnetic impurity is realized as a spin-selective $\delta$-potential in the center of the trap. Like in the many-body case, the attractive interaction between particles favors pair formation, which leads to a few-body analogue of an energy gap [12, 25]. The interplay between the time-reversal-symmetric pair formation and the magnetic impurity then drives the few-body analogue of the QPT. For strong impurity-fermion couplings, it is energetically favorable to shield the impurity via forming a "bound" state with one of the particles, thereby changing the fermionic parity of the ground state. We remark that both the impurity-fermion and fermion-fermion interactions are essential here. This is in stark contrast to the physics driven by a global magnetic field, which breaks pairs due to energetically mismatched Fermi surfaces, and leads to a level crossing even at vanishing fermion-fermion interactions.

While ab-initio results are not possible in the many-body limit, which usually requires mean-field and $T$-matrix approximations with varying degrees of self-consistency [19, 20], our

few-body model can be studied in a numerically exact manner. As a numerical method, we choose the full-configuration interaction (FCI) method in a truncated model space (also referred to as exact diagonalization). To avoid exorbitant numerical cost that occurs for overly large model spaces, we employ an effective two-body interaction in a truncated space, which is inspired by the Lee-Suzuki method known in nuclear physics, see Ref. [26] for review. One can find a description of the effective interaction for cold-atom systems in Ref. [27]; its applications for particles in one-dimensional harmonic traps and rings are discussed in Refs. [28–30]. Our FCI code, written in the Julia language, along with a detailed description of the method as well as an extensive benchmark is available in Ref. [31].

As an alternative numerical approach we use the transcorrelated method (TCM) [32,33] to generate benchmark data for validating the effective interaction results. The TCM removes the wave function cusp with a similarity transformation of the Hamiltonian, which improves the convergence properties in a finite basis set expansion. The transcorrelated method for short-range interactions was described and benchmarked for few-fermion systems in one spatial dimension in Ref. [34] and in three dimensions in Ref. [35].

The remainder of this work is organized as follows: In Sect. 2, we present the model of interest. We also briefly discuss the numerical approaches. We proceed with the main section of this work in Sect. 3, where we first review the emergence of pairing in 1D traps and subsequently investigate the effect of a magnetic impurity. We relegate the discussion of technical details as well as the discussion of additional data to Apps. A and B, respectively. Furthermore, we calculate experimentally accessible two-body correlation functions that exhibit signatures of the underlying physics. Finally, we summarize our work and propose an experimental realization in Sect. 4.

## 2    Model and numerical methods

We are interested in an ensemble of a few harmonically trapped two-component fermions, described by the one-dimensional Hamiltonian

$$H = \sum_{i=1}^{N_\uparrow} \left( -\frac{\hbar^2}{2m}\nabla_{x_i}^2 + \frac{m\omega^2}{2}x_i^2 \right) + \sum_{j=1}^{N_\downarrow} \left( -\frac{\hbar^2}{2m}\nabla_{y_j}^2 + \frac{m\omega^2}{2}y_j^2 \right) + g\sum_{i,j}\delta(x_i - y_j) + H_{\text{imp}}, \quad (1)$$

where $N_\uparrow$ ($N_\downarrow$) is the number of spin-up (spin-down) fermions, $\omega$ is the frequency of the trap, $m$ is the particle mass and $g$ is the strength of interaction. $x_i(y_j)$ is the position of the $i$th spin-up ($j$th spin-down) fermion. Our focus is on attractively interacting particles, i.e., $g < 0$, since we are interested in the physics associated with pairing of fermions. Note that the spin projection of a particle is fixed – a standard assumption for cold-atom systems. The present study is limited to systems with $N_\uparrow + N_\downarrow \leq 9$, which can be reliably addressed using our implementation of exact diagonalization.

The term $H_{\text{imp}}$ contains a spin-dependent external potential, which we use to model a magnetic impurity. For simplicity, we employ a spin-selective $\delta$-potential in the center of the trap such that

$$H_{\text{imp}} = G_\uparrow \sum_{i=1}^{N_\uparrow} \delta(x_i) + G_\downarrow \sum_{j=1}^{N_\downarrow} \delta(y_j), \quad (2)$$

where $G_\sigma$ encodes properties of the fermion-impurity scattering. It is worth pointing out that the shape of the fermion-impurity potential is not of great importance as long as the width of this potential is smaller than any other relevant length scales of the problem, which are described below. In the present work, for simplicity, we use the convention $G_\uparrow = -G_\downarrow \equiv G > 0$ so that $\uparrow$ particles are repelled and $\downarrow$ particles attracted by the central impurity with equal magnitude. Our results will also qualitatively describe the situation $G_\uparrow \neq -G_\downarrow$ as long as the impurity attracts only one spin-type of the fermions so that a bound-state may form.

*Length Scales of the Problem:* The characteristic length scale associated with the fermion-fermion interaction in Eq. (1) is given by the one-dimensional scattering length, $a_0$, which is defined as $a_0 = -2\hbar^2/mg$, see, e.g., [6]. Similarly, one can define a length scale associated with the fermion-impurity interaction. The characteristic length scale for the trap is given by the harmonic oscillator length $\xi = \sqrt{\hbar/m\omega}$. For the remainder of this work, we shall use the harmonic oscillator units in which $\xi = 1$ and $\hbar\omega = 1$. The fourth relevant length scale is connected to the Fermi momentum, see also subsection 3.3.

## 2.1 Discussion of the model

Before proceeding with our analysis, it is worthwhile to motivate the choice of the Hamiltonian in Eq. (1). To this end, we first recall the fact that one-dimensional Fermi gases with short-range attractive interactions and $H_{\text{imp}} = 0$ exhibit an $s$-wave pairing gap already at the few-body level[1]. For the harmonically trapped 1D Fermi gases, this has been discussed in Refs. [25, 36] (see also subsection 3.1).

If $H_{\text{imp}} \neq 0$, our system includes a spin-dependent external potential which models a magnetic impurity. This allows us to study the interplay between locally broken time-reversal symmetry and $s$-wave pairing. For $N_\uparrow, N_\downarrow \to \infty$, it is known that a magnetic impurity in the vicinity of an $s$-wave superconductor leads to the so-called Shiba states in the excitation spectrum. These are energy levels within the pairing gap induced by the local breaking of time-reversal symmetry [19, 20]. These sub-gap states are conventionally studied in the grand-canonical ensemble where they emerge as a pair of excitations symmetrically around the chemical potential.

In the present work, we study few-body systems with a well-defined number of particles, therefore, we are not able to see this behavior explicitly. However, we can study a few-body analogue of this physics by diagonalizing the Hamiltonian in two different sectors which differ in the fermion parity, namely in the $(N_\uparrow, N_\downarrow) = (N, N)$ and $(N, N + 1)$ sectors referred to as $S = 0$ and $S = \frac{1}{2}$, respectively. Without the impurity, i.e., if $G = 0$, the ground state should be in the $S = 0$ sector, where the pairing is strongest. However, if the fermion-impurity interaction is strong enough to break a pair, then the $S = \frac{1}{2}$ sector may host the lowest energy level. This precursor of a QPT is sketched in Fig. 1, where also the bound state of one of the excess particles and the impurity is indicated as the reason for the energetically favorable configuration. It is reasonable to expect that for a large enough number of particles our model reproduces the behavior in the grand-canonical picture. Meanwhile, at small number of particles, our model offers a systematic way to approach a QPT (amenable also to analogue quantum simulation) as we shall discuss in the following.

---

[1]More precisely, we mean that correlations in the attractive few-body system feature a precursor to the $s$-wave pairing gap. The emergence of the BCS (Bardeen-Cooper-Schrieffer) pairing is technically constrained to the many-body limit, where (typically) particle conservation is not assumed.

## 2.2 Numerical methods

To find the spectrum of the Hamiltonian, we diagonalize the Hamiltonian in a truncated Hilbert space. Our main tool is an effective interaction approach in the harmonic oscillator basis, which we use to obtain our desired few-body phase diagrams. Additionally, we validate our results without the magnetic impurity with the transcorrelated method. Both approaches are briefly described below.

### 2.2.1 Effective interaction approach

To truncate the Hilbert space with the effective interaction approach, we keep only $n_b$ lowest eigenstates of the harmonic oscillator basis. While the truncation is a necessary step to make the problem amenable to numerical treatment, it introduces a bias due to the discarded physical states. To mitigate this shortcoming, an extrapolation to the infinite-basis limit is required. However, the numerical cost for precise extrapolation rises combinatorially and therefore becomes problematic with more than very few particles. In order to minimize the numerical effort – while still maintaining accuracy – we make use of an effective interaction approach known in the nuclear-physics community in the context of the no-core shell model [27]. The key step is to replace the bare two-body matrix elements with effective values optimized for the applied truncation scheme. This step essentially constitutes a particularly effective renormalization procedure with regard to the two-body problem: Instead of fixing only a single parameter (the coupling strength $g$), the effective-interaction approach amounts to tuning all interaction matrix elements in order to match the lowest part of the energy spectrum to the analytic solution. This can be achieved by constructing an effective interaction from the effective Hamiltonian whose matrix representation reads as follows

$$H^{\text{eff}} \equiv U^{\dagger}\text{diag}(E_1, ..., E_n)U \,, \tag{3}$$

where $E_1, ..., E_n$ are the $n$ lowest two-body eigenenergies, which can be calculated exactly [37, 38]; $U$ is a matrix whose rows are formed by the corresponding eigenvectors projected on the truncated Hilbert space. The effective potential $V^{\text{eff}} = H^{\text{eff}} - T$ ($T$ is the kinetic energy operator) exactly reproduces the infinite-basis spectrum for the two-body problem already at a finite basis cutoff. Moreover, and this is the crucial numerical benefit, the effective interaction substantially improves the convergence properties of FCI calculations for $N > 2$. Consequently, numerical values obtained in small truncated Hilbert spaces may be much more accurate than those obtained for a bare interaction. The significantly reduced numerical effort allows us not only to probe larger systems reliably but also to scan cheaply the parameter space and thus map out few-body phase-diagrams. For details of the method as well as on our implementation we refer to Ref. [31].

### 2.2.2 Transcorrelated Method

To have additional benchmark data for the effective interaction approach, we use a potentially more accurate, but more expensive TCM for short-range interactions [34, 35]. The $\delta$-function interaction, equivalent to the Bethe-Peierls (BP) boundary condition, produces a cusp in the wave function, which is difficult to capture with a basis set expansion. To mitigate this problem, we introduce a Jastrow factor $e^{\tau}$, where $\tau$ is a function of all the particle coordinates $x_i$ and $y_i$. The Jastrow factor includes the cusp that satisfies the BP boundary conditions,

and is folded into the Hamiltonian with a similarity transformation

$$\tilde{H} = e^{-\tau} H e^{\tau}.$$

This removes the cusp from the wave function, which greatly improves the convergence with respect to a basis set expansion of the transcorrelated Hamiltonian $\tilde{H}$. In particular, for interacting spin-$\frac{1}{2}$ fermions in one dimension, the convergence of the ground-state energy improves from $n_b^{-1}$ to $n_b^{-3}$ when expanding in a truncated single-particle basis with $n_b$ plane waves [34].

The downside of this approach is that it makes the Hamiltonian $\tilde{H}$ non-Hermitian and more complicated to construct. Still, it can be diagonalized with the widely available Arnoldi iteration, or in the case of larger systems, with full-configuration interaction quantum Monte Carlo (FCIQMC) [39,40].

For the results reported in this work we tightly fit a box of length $L$ in real-space to capture the relevant parts of the ground state wave function. We then use a single-particle basis with $n_b$ plane waves to expand $\tilde{H}$, find the ground state energy, and increase $L$ and $n_b$ until the desired accuracy is reached. The results in this paper were produced with our Julia package `Rimu.jl` [41], which includes implementations of the transcorrelated Hamiltonian construction and FCIQMC.

## 3 Results

In this section, we present our central results that concern few-body systems with a magnetic impurity.

### 3.1 Balanced systems without a magnetic impurity

Before addressing systems with a magnetic impurity, let us consider a balanced system $N_\uparrow = N_\downarrow$ with $G = 0$. Our aim here is to highlight the precursor of a pairing gap in the few-body limit, see also the discussion in Ref. [25]. With $G = 0$, the Hamiltonian is symmetric with respect to an exchange of spin up and spin down particles. Indeed, it is clear that the exchange $\{x_i\} \rightarrow \{y_i\}$ does not change the Hamiltonian in Eq. (1). This implies that if $\psi(x_1, ..., x_{N_\downarrow}; y_1, ..., y_{N_\uparrow})$ is an eigenstate of $H$, then $\psi(y_1, ..., y_{N_\uparrow}; x_1, ..., x_{N_\downarrow})$ is also an eigenstate of $H$.

Let us introduce the swap operator $\mathcal{T}$, which performs the transformation $\{x_i\} \rightarrow \{y_i\}$. Since $\mathcal{T}$ and $H$ commute, then every eigenstate of $\mathcal{H}$ can be labeled using the eigenvalues $T = \pm 1$ of $\mathcal{T}$. It is worth noting that in the spin language, the operator $\mathcal{T}$ is the spin-flip operator, which enters the time-reversal operator. In the limit $N_\uparrow, N_\downarrow \rightarrow \infty$, the system features a spin gap (see, e.g., [42]), i.e., there is an energy difference between the singlet ground state and the first triplet excited state. This gap can be connected to the energy difference between the manifolds with $T = 1$ and $T = -1$, see Ref. [25] for a more detailed discussion.

Let us illustrate the 'spin-flip' symmetry for the simplest balanced system, i.e., for $1 \uparrow +1 \downarrow$, which can be solved analytically [37, 38]. The wave function for this system is written as

$$\psi(x_1, y_1) = \phi_{\text{cm}}(x_1 + y_1)\, \phi_{\text{rel}}(x_1 - y_1), \tag{4}$$

where $\phi_{\text{cm}}$ and $\phi_{\text{rel}}$ describe the center-of-mass and relative motion, respectively. The center-of-mass part is always symmetric with respect to an exchange of particles. In the relative

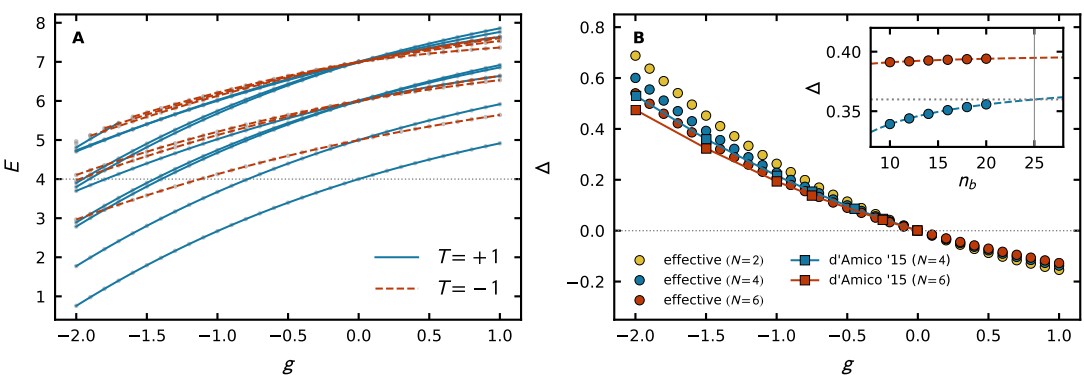

Figure 2: **Spectrum and pairing gap.** (A) The spectrum of the $2\uparrow+2\downarrow$ system as a function of the interaction strength $g$. The (blue) solid curves show the states with $T=1$; the (red) dashed curves show the states with $T=-1$. The difference between the lowest dashed curve and the lowest solid curve is written as $2\Delta+1$, where $\Delta$ is the pairing gap. (B) The pairing gap as a function of the interaction strength $g$. We also present the data from an earlier FCI calculation with bare interaction (squares, lines are added to guide an eye) [25]. (Inset) Convergence of the gap as a function of the basis cutoff $n_b$ for bare (blue symbols) and effective (red symbols) interactions for $N=2\uparrow+2\downarrow$ at $g=-1.5$, including a fit to the respective data (dashed curves). The horizontal dotted line is the result from [25] obtained with $n_b=25$. Our data in the main plot are obtained with $n_b=13$, for which the error-bars are on the scale of the corresponding symbols.

part, the exchange of particles corresponds to a parity operation. The even functions $\phi_{\rm rel}$ correspond to $T=1$; odd functions have $T=-1$. For $g<0$ there is always an energy difference between the lowest states in the $T=1$ and $T=-1$ manifolds, which we write as $2\Delta+1$. The parameter $\Delta$ here can be linked to a few-body analogue of the pairing gap that appears in a many-body Fermi system, see Ref. [25] for a more detailed explanation[2]. It is worth noting that the discussion above applies also to systems without a trap, in which case the energy difference between the manifolds is simply given by the two-body binding energy.

We illustrate the few-body analogue of the pairing gap for $N_\uparrow+N_\downarrow=2,4,6$ in Fig. 2. In panel A, we show the low-lying energy spectrum of the $2\uparrow+2\downarrow$ system as a function of the interaction strength $g$ resolved for both the positive (solid blue curves) and negative (dashed red curves) symmetry sectors of the swap operator $\mathcal{T}$. In panel B, we show the pairing gap, which is determined from the level spacing between the lowest states in each respective sector. We compare these results to the values of Ref. [25] and report a good agreement for all available interaction strengths, modulo a slight deviation at larger coupling, which is likely an artefact of the convergence with the bare interaction (see the inset of Fig. 2). The results obtained with the effective interaction have been computed by maximally considering $n_b=13$ single-particle states which was enough to observe convergence essentially indiscernible at the scale of the figure. For the present work it is exactly the emergence of a positive few-body gap for $g<0$ that allows for the study of a precursor of a QPT due to a non-trivial interplay between particle pairing and impurity scattering, which we will discuss in the following.

---

[2]In short, the additive term 1 in $2\Delta+1$ accounts for the spacing between energy levels of a non-interacting system; the factor 2 in front of $\Delta$ appears because the spin flip leaves two atoms unpaired – not a single atom as in the standard definition of the gap.

It is worth noting that the existence of a negative "gap" ($\Delta < 0$) on the repulsive side is a mere consequence of the finite number of particles in the trap – excitations between even and odd spin-flip symmetry should be available at no energy cost in the thermodynamic limit. For the emergence of non-analytic behavior, i.e., an expected quantum phase transition at $g = 0$, one would have to carefully extrapolate to an infinite number of particles, where the lowest states of both sectors become degenerate. Such an extrapolation, however, is far from trivial and certainly beyond the scope of the present work or any exact diagonalization study (see, e.g., Ref. [12] for a discussion of the matter).

## 3.2 Two- and three-body systems with a magnetic impurity

To provide a basic understanding about the role of the magnetic impurity, we discuss the systems $1 \uparrow +1 \downarrow$ and $1 \uparrow +2 \downarrow$ in this subsection. In general, these systems are not solvable analytically (see, e.g., [43]), and one has to rely on numerical methods. Still, as we show below, the limiting cases $G \to \infty$ and $g \to -\infty$ can be addressed, providing some additional insight into the problem. In particular, they show that the ground-state energy of the 'spin triplet' $1 \uparrow +2 \downarrow$ system can be lower than the ground-state energy of the 'spin singlet' $1 \uparrow +1 \downarrow$ system only if both $G$ and $|g|$ are sufficiently large.

### 3.2.1 Limiting cases: analytic insight into strong coupling

*Limit $G \to \infty$:* We first consider the $1 \uparrow +1 \downarrow$ system with strong fermion-impurity interactions. If $g = 0$, then the Hamiltonian in Eq. (1) does not couple spin-up and spin-down particles. Therefore, the energy of $1 \uparrow +1 \downarrow$ equals the energy of $0 \uparrow +1 \downarrow$ plus the energy of $1 \uparrow +0 \downarrow$. In the $0 \uparrow +1 \downarrow$ system, the fermion and the impurity form a tightly-bound state, whose energy is $\epsilon_G$ and can be calculated analytically as in Refs. [37, 38]. Note that for large values of $G$, the harmonic trap plays a minor role and $\epsilon_G \simeq -G^2/2$ in agreement with the textbook calculations. In the $1 \uparrow +0 \downarrow$ system, the fermion feels an impenetrable wall in the middle of the trap. The corresponding wave function must vanish at $x_1 = 0$, which means that the ground-state energy is equal to that of the first excited state of the harmonic oscillator, i.e., to $3/2$. Therefore, the energy of $1 \uparrow +1 \downarrow$ is $\epsilon_G + 3/2$. Spin-up and spin-down particles have no overlap if $G \to \infty$, which implies that finite values of $g$ do not change the energy of the system.

For the $1 \uparrow +2 \downarrow$ system, the calculations are more complicated. For $g = 0$, the ground-state energy can be calculated by considering the systems $1 \uparrow +0 \downarrow$ and $0 \uparrow +2 \downarrow$ separately. We obtain $\epsilon_G + 3$, which is larger than the energy of the $1 \uparrow +1 \downarrow$ system. However, as we show below, there is a value of $g$ for which the energy of $1 \uparrow +2 \downarrow$ is equal to the energy of $1 \uparrow +1 \downarrow$. This critical value of $g$ is of our interest here.

For $g \neq 0$, the $1 \uparrow +2 \downarrow$ system can be effectively described using an auxiliary $1 \uparrow +1 \downarrow$ problem in a harmonic trap with an impenetrable wall in the middle for both spins. The spin-up fermion feels a wall due to the condition $G \to \infty$. The spin-down fermion cannot go to the origin due the second spin-down fermion already attracted by the magnetic impurity. The auxiliary problem is described by the Hamiltonian

$$H_A = -\frac{1}{2}\frac{\partial^2}{\partial x_1^2} - \frac{1}{2}\frac{\partial^2}{\partial y_1^2} + \frac{x_1^2}{2} + \frac{y_1^2}{2} + g\delta(x_1 - y_1), \tag{5}$$

with the boundary condition $\psi(x_1 = 0, y_1) = \psi(x_1, y_1 = 0) = 0$. For $g > 0$, the Hamiltonian $H_A$ was studied in Ref. [43]. Here, we are interested in the case with $g < 0$.

To provide some analytical insight into the problem, we aim to find an approximate value to the energy using a variational ansatz for the Hamiltonian in the polar coordinates ($x_1 = r\cos\phi, y_1 = r\sin\phi$):

$$H_A = -\frac{1}{2}\frac{\partial^2}{\partial r^2} - \frac{1}{2r}\frac{\partial}{\partial r} - \frac{1}{2r^2}\frac{\partial^2}{\partial\phi^2} + \frac{r^2}{2} + \frac{g}{\sqrt{2}r}\delta(\phi - \pi/4), \tag{6}$$

where for the ground state we shall only consider $0 < \phi < \pi/2$, since the wave function vanishes at the boundaries, i.e., at $\phi = 0$ and $\phi = \pi/2$. A suitable variational function reads as

$$f = Ae^{-r^2/2}r^\mu F(\phi), \tag{7}$$

where $A$ is the normalization constant, $\mu$ is the variational parameter, and the function $F$ has the form

$$F(\phi) = \begin{cases} \sin(\mu\phi) & \text{if } \phi \in [0, \pi/4], \\ \sin(\mu(\pi/2 - \phi)) & \text{if } \phi \in (\pi/4, \pi/2]. \end{cases} \tag{8}$$

It satisfies the boundary condition $F(0) = F(\pi/2) = 0$ by construction.

A few comments about the variational function in Eq. (7) are in order here. With $\mu = 2$, the function $f$ solves the problem at $g = 0$. For other values of $\mu$, the function accounts for the singularity due the delta-function interaction. Our choice of $F$ is motivated only for small values of $g$. For larger values, a function that more faithfully represents a bound state might be needed. For a more detailed discussion on the physics of the employed variational ansatz, we refer to Ref. [44].

For the variational function from Eq. 7, the expectation value of $H_A$ is

$$\langle f|H_A|f\rangle = (1+\mu) + \left(\mu\sin\left(\frac{\mu\pi}{2}\right)\frac{\Gamma[\mu]}{\Gamma[1+\mu]} + g\sqrt{2}\sin^2\left(\frac{\mu\pi}{4}\right)\frac{\Gamma[\mu+\frac{1}{2}]}{\Gamma[1+\mu]}\right)\frac{2\mu}{\pi\mu - 2\sin\left(\frac{\mu\pi}{2}\right)}, \tag{9}$$

where $\Gamma$ is the Gamma function. We minimize $\langle f|H_A|f\rangle$ with respect to $\mu$, and obtain an approximation to the ground-state energy. The corresponding approximation to the energy of the $1\uparrow + 2\downarrow$ system is $\epsilon_G + \langle f|H_A|f\rangle$. For $g = 0$, the minimum of $\langle f|H_A|f\rangle$ equals 3, and it is reached for $\mu = 2$, as expected. For $g \simeq -1.95$, the minimum of $\langle f|H_A|f\rangle$ equals 1.5, the corresponding value of $\mu$ is approximately 1.3.

To summarize, for $g \gtrsim -1.95$, the Pauli pressure makes the energy of the $1\uparrow + 2\downarrow$ system higher than the energy of the $1\uparrow + 1\downarrow$ system. For $g \lesssim -1.95$, the fermion-fermion pairing makes the $1\uparrow + 2\downarrow$ system energetically more favorable than $1\uparrow + 1\downarrow$. Although, the critical value is obtained here using a number of approximations, we will show in the next subsection that it is accurate by comparing to the numerical results based upon exact diagonalization.

*Limit $g \to -\infty$:* Here, we consider the system with strong fermion-fermion interaction. The $1\uparrow + 1\downarrow$ system with $G = 0$ was investigated in Refs. [37, 38]. It is most easily solved by decoupling the relative motion from the center-of-mass coordinate, see Eq. (4). The interaction enters only in the former part; it leads to a formation of a tightly bound dimer, whose energy is $\epsilon_g$. The (total) ground-state energy of the $1\uparrow + 1\downarrow$ system is $\epsilon_g + 1/2$, where $1/2$ is the zero-point energy of the center-of-mass Hamiltonian. For the $1\uparrow + 2\downarrow$ system, the bound state becomes transparent to the extra fermion [45]. Its energy is thus $\epsilon_g + 1$.

If we turn on $G$, then the net effect of the perturbing potential, $G\delta(x_1) - G\delta(y_1)$, on the dimer of the $1\uparrow + 1\downarrow$ system is zero. Indeed, the dimer is tightly bound, and whenever the potential $G\delta(x_1)$ acts on a spin-up particle, the potential $-G\delta(y_1)$ acts on a spin down

particle. This means that $\epsilon_g + 1/2$ is an accurate approximation to the energy of the $1 \uparrow +1 \downarrow$ system also for finite values of $G$.

To investigate the $1 \uparrow +2 \downarrow$ system, we can make use of the previously mentioned transparency of the strongly-bound dimer to an extra fermion. Therefore, the energies of $1 \uparrow +1 \downarrow$ and $1 \uparrow +2 \downarrow$ are equal when the energy of the $0 \uparrow +1 \downarrow$ system vanishes, which happens at (see Refs. [37, 38])

$$G = 2\frac{\Gamma[3/2]}{\Gamma[1/4]} \simeq 0.5. \qquad (10)$$

All in all, the considered limiting cases suggest a curve in parameter space $g - G$ that separates the $S = 0$ from $S = \frac{1}{2}$ sectors. We use numerical methods to find this curve in the following subsection.

### 3.2.2 Numerical results

Here, we discuss our numerical results for small systems with general couplings $g$ and $G$. To analyze the potential precursor of a many-body QPT, we focus on the energy difference $\Delta_{21} = E(1 \uparrow +2 \downarrow) - E(1 \uparrow +1 \downarrow)$ which features a sign change when the parity of the ground-state changes.

Unless otherwise noted, we discuss results of the exact diagonalization method with effective potential, see 2.2.1. The presented ground-state energies are obtained by extrapolating to the infinite-basis limit $n_b \to \infty$ according to the functional form (see also App. A)

$$E(n_b) = E_\infty + \frac{a}{n_b^\sigma}. \qquad (11)$$

Here, $E_\infty$ denotes the extrapolated ground-state energy; $\sigma$ is the exponent of the convergence. While $\sigma$ was found to be 0.5 for the conventional bare interaction [36], we empirically found that $\sigma = 1.0$ yields the most accurate results for the effective interaction approach used in this study. To visualize the size of the extrapolation effect, we include results of different extrapolation schemes parametrized by $\sigma$ where applicable.

We present $\Delta_{21}$ for $g = 0$ in panel A of Fig. 3. The constant value of $\Delta_{21}$ may be straightforwardly derived since for $g = 0$ the system is a collection of non-interacting fermions whose energy is a sum of one-body energies. The value of $\Delta_{21}$ is given by the spin-down particle in the first excited state of the harmonic oscillator. This energy equals $3/2$. It is independent of $G$ [37, 38], due to the odd parity (and hence a node in the trap center) of the first excited state.

In panel B of the same figure, we present $\Delta_{21}$ for the the opposite case, namely $G = 0$ (i.e., without the magnetic impurity) as a function of the particle interaction strength $g$. Since there is no breaking of spin-flip symmetry one would expect no crossover as a function of interaction strength, so that the balanced $1 \uparrow +1 \downarrow$ case remains the ground state for all couplings. Moreover, as pointed out above, at $g \to -\infty$ the dimer is transparent for the fermion [45], therefore, $\Delta_{21}$ is expected to converge to $1/2$ in the limit $|g| \to \infty$.

As apparent from the figure, these expectations are supported by our numerical data. Note, that the current implementation of the effective interaction approach for this configuration is constrained to the (already substantial) interaction strength $|g| \lesssim 5$, beyond which the discrepancy between the various extrapolation schemes becomes sizeable (this area is marked in gray in the figure) and eventually are too contaminated by finite-size effects to allow for reliable statements. Results from the TCM at fixed $n_b = 41$ are shown in the same panel

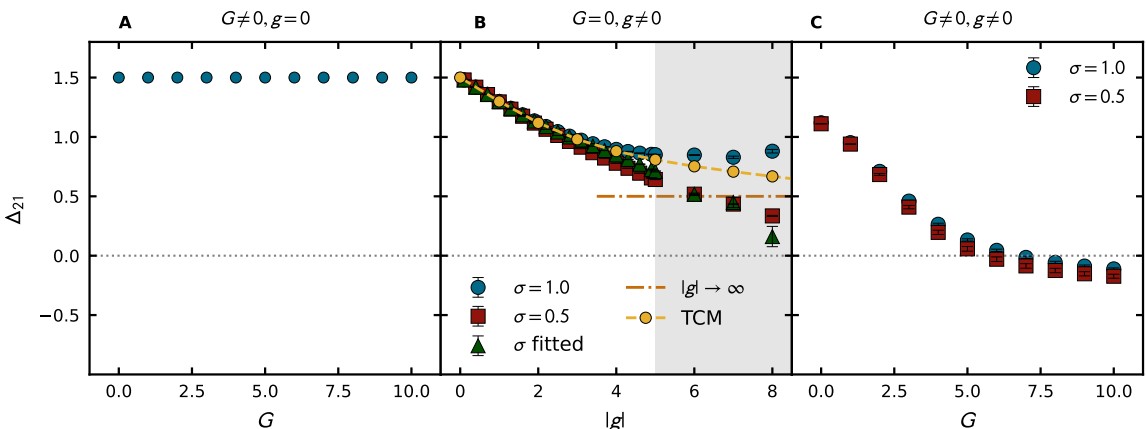

Figure 3: **Few-body systems with a magnetic impurity.** Panels show the difference between the ground-state energies of the $1\uparrow+2\downarrow$ and $1\uparrow+1\downarrow$ systems, $\Delta_{21}$. (A) Results for vanishing fermion-fermion interactions, $g=0$. (B) Results for $G=0$, i.e., without a magnetic impurity. Here, different markers correspond to three extrapolation schemes (see Eq. 11 and App. A). The error bars show the associated fitting error. The shaded area marks the parameter region, where the result strongly depends on the extrapolation scheme. This parameter region shall not be considered later. The yellow disks are the results of the transcorrelated method (TCM); the corresponding dashed curve is added to guide the eye. The orange dashed-dotted line is our analytical prediction for the limit $g \rightarrow -\infty$. (C) Generic case where $G \neq 0$ and $g \neq 0$ (here, $g=-2.0$). We interpret the change in the sign of $\Delta_{21}$ as a few-body precursor of a transition from the $S=0$ to $S=\frac{1}{2}$ ground state.

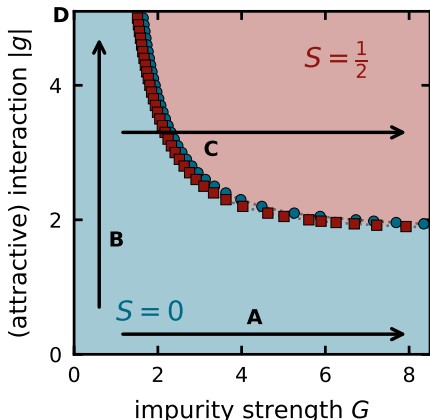

Figure 4: **Few-body "phase diagram" with a magnetic impurity in the $g - G$ plane**. Different markers correspond to different fit functions as in panel B of Fig. 3. The three arrows schematically show the three cases discussed in panels A, B and C Fig. 3.

with yellow symbols (and dashed line). These results are available for the case without the magnetic impurity, and show a smooth convergence to the expected limiting value $1/2$ for $|g| \to \infty$. The TCM data validate the effective interaction approach and our interpolation scheme (see Eq. (11)) with fixed exponent $\sigma = 1$.

Finally, in panel C of Fig. 3, we illustrate the generic case when both $g$ and $G$ are finite. The gap, $\Delta_{21}$, is shown as a function of $G$ at fixed interaction strength $g = -2$. We observe that a ground-state level crossing occurs when $\Delta_{21} = 0$ (indicated by the dotted gray line). For large values of $G$, the fermion-fermion pairing pulls the energy of the $1 \uparrow + 2 \downarrow$ system below that of the balanced system. Excitations between the two sectors, which involve the change of the fermionic parity (or, equivalently, a change of particle number), can be considered as a few-body counterpart of the sub-gap states that occur in the many-body limit. For completeness, we show results of different extrapolation schemes in Fig. 3 C. Note that the point where $\Delta_{21}$ vanishes is (almost) independent of the extrapolation scheme for all considered couplings, i.e., couplings outside the shaded area in panel Fig. 3 B.

We combine the above findings to produce the 'few-body phase diagram' in the $g$ vs. $G$ parameter plane, see Fig. 4. The curve in the figure is determined by the condition $\Delta_{21} = 0$. In the area '$S = \frac{1}{2}$', the ground-state energy of $1 \uparrow + 2 \downarrow$ is below that of $1 \uparrow + 1 \downarrow$. The opposite is true otherwise. The three possible scenarios illustrated in panels A, B, and C of Fig. 3 are indicated by the black arrows, where the '$x$'-direction of the associated panel agrees with the direction of the arrows. From this analysis it indeed becomes apparent that both couplings $g$ and $G$ need to be substantial in order to enable a ground-state transition and, hence, a few-body analogue of a QPT. To estimate a potential systematic error, we show our data corresponding to $\sigma = 0.5$ and $1$ (same color scheme as in Fig. 3), see also App. B. The phase diagram beyond the presented interactions $g$ and $G$ requires more involved numerical analysis. However, it will not change the takeaway message conveyed here.

Finally, we note that a simple energy shift due to a global magnetic field (described by $B\sigma_z$) may also lead to $\Delta_{21} < 0$, however, the underlying physics would be very different. Indeed, a global magnetic field would lead to a crossing of the energy levels regardless of the

attractive interaction strength. Only a local magnetic field that we study here may lead to a subtle interplay between pair formation and spin-dependent scattering off the impurity.

## 3.3 Approaching the many-body limit: scaling with particle number

So far we have investigated the effect of the magnetic impurity in the smallest possible system and mapped out the 'few-body phase-diagram' in the $g-G$ plane. In this section, we calculate 'few-body phase-diagrams' for larger systems (see also App. B), which allow us to study the crossover from few- to many-body physics in the presence of a magnetic impurity.

*Units.* For a meaningful analysis, here, we need to change the size of the system along with the number of particles[3]. For a harmonic trap potential, it makes sense to change the frequency of the trap such that the density in the middle of the trap is fixed, $\rho = k_F/\pi$, where[4] $k_F = \sqrt{(2N_\uparrow - 1)m\omega/\hbar}$ denotes the wave vector associated with the Fermi energy $E_F = \hbar^2 k_F^2/(2m)$, see, e.g., Ref. [46]. Such a change of the trap should allow for a faithful comparison of physics due to the impurity in the middle of the trap (see also Refs. [9,17] for a relevant discussion of mobile impurities in a Fermi gas).

In practice, we fix $\omega$, and study the few- to many-body transition by rescaling the couplings with the corresponding $k_F$

$$\tilde{g} = \frac{g}{\sqrt{2N_\uparrow - 1}}, \qquad \tilde{G} = \frac{G}{\sqrt{2N_\uparrow - 1}}. \tag{12}$$

For simplicity, we perform rescaling only for a balanced system, i.e., for $N_\uparrow = N_\downarrow$. For an imbalanced system ($N_\downarrow = N_\uparrow + 1$), we use $\tilde{g}$ and $\tilde{G}$ defined by $N_\uparrow$.

*Limiting cases.* We can analyze the limiting case $g \to -\infty$ following the discussion in 3.2.1. The limit $G \to \infty$ requires more involved calculations, and we leave its investigation to future studies.

For $g \to -\infty$, a spin-up and a spin-down fermion form a dimer. The balanced system is then equivalent to the Tonks-Girardeau gas, see, e.g., Ref. [47]. The $N_\downarrow = N_\uparrow + 1$ system has an additional fermion, which does not interact with the Tonks-Girardeau gas. The absence of fermion-dimer interaction means that the critical value of $G$ is independent of the size of the system. Therefore, we can use the result of Eq. (10). In the rescaled units, this equation can be written as

$$\tilde{G} \simeq \frac{0.5}{\sqrt{2N_\uparrow - 1}}. \tag{13}$$

Note that this value vanishes for the many-body system ($N_\uparrow \to \infty$).

*Numerical analysis.* We extract 'few-body phase-diagrams' from the condition on the ground state energies: $E(N\uparrow + N\downarrow) = E(N\uparrow + (N+1)\downarrow)$. Fig. 5 presents our findings for systems with up to $N_\uparrow = 4$, i.e., for systems as large as $4\uparrow + 5\downarrow$. We note that the transition points for $N_\uparrow = 1$ stand out in comparison to the ones for higher particle numbers – this is likely an artefact of our system of units which merely fixes a scale relevant in the many-body limit. For larger particle numbers, although not fully "converged-to-many-body-limit", the data obtained with extrapolated ground-state energies seem to collapse on a curve, which is almost independent on $N_\uparrow$, as shown in panel A of Fig. 5. For example, the data for the

---

[3]In a box potential of length $L$, one would typically change the size of the system such that the density, e.g., $\rho = N_\uparrow/L$, is kept constant.

[4]Here, we use dimensionful quantities, for clarity.

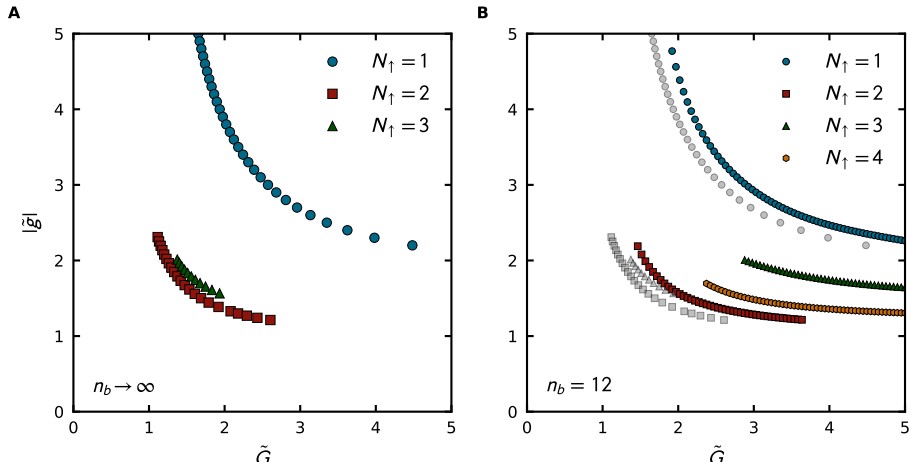

Figure 5: **Approaching the many-body limit.** Points where the ground-state energies for balanced and imbalanced particle configurations are equal, i.e., where $E(N\uparrow+N\downarrow)=E(N\uparrow+(N+1)\downarrow)$. The data are shown as a function of the rescaled fermion-fermion interaction strength $\tilde{g}$ and the rescaled magnetic impurity interaction strength $\tilde{G}$, see Eq. (12). Results are shown for $N_\uparrow \leq 4$, see the legend. In all cases the upper left (lower right) area corresponds to the $S=1/2$ ($S=0$) phase. (A) Phase diagrams obtained from ground-state energies extrapolated to the infinite basis limit (B) Phase diagrams at finite cutoff $n_b = 12$ (colored symbols) compared to the extrapolated results (gray symbols).

systems with $N_\uparrow = 2$ and $N_\uparrow = 3$ are very close. Our interpretation is that the impurity is screened by only a few fermions. Thus, including more particles, thereby increasing the size of the system, cannot have a strong effect. Similar behavior was also observed in other impurity systems, see for example [17].

Finally, we comment on the accuracy of the presented data. The maximally attainable cutoff value $n_b$ is reduced when we increase the particle number. However, we can reach sufficient convergence across a range of couplings (see also [31]). To be specific, for $N_\uparrow = 1$ we compute data using up to $n_b = 40$ states in the single-particle basis while for $N_\uparrow = 4$ we use only $n_b = 8$ to 13 one-body states[5]. Our values up to systems of size $3\uparrow+4\downarrow$ are well-converged and extrapolation is under control. For larger systems, the exact values may shift slightly. However, the obtained accuracy is enough for a qualitative discussion. This point is further addressed in panel B of Fig. 5, where we show the transition points at finite cutoff $n_b = 12$ for all particle configurations. The panel also shows clustering of the data, which allows us to conclude that a more accurate determination of the ground-state energies will not change the general behavior of the phase-diagram.

## 3.4 Other observables

To this moment, we characterized the few-body systems only by their energies. This allowed us to show similarities between our model and a magnetic impurity in the vicinity of

---

[5]For the $4\uparrow+5\downarrow$ system, the cutoff value $n_b = 13$ leads to a dimension of the Hilbert space, $\dim\mathcal{H} = 920205$, which is at the limit of our numerical approach.

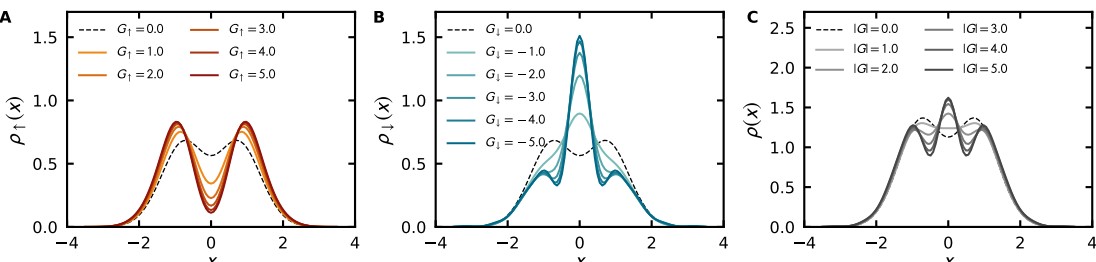

Figure 6: **Density profiles.** The figure illustrates the density profiles for the $2\uparrow +2\downarrow$ system without fermion-fermion interactions (i.e., with $g=0$) for representative values of $G$, see the legend. (A) Density of spin-up particles, $\rho_\uparrow$. (B) Density of spin-down particles, $\rho_\downarrow$. (C) Total density, $\rho = \rho_\uparrow + \rho_\downarrow$. In all panels the black dashed line corresponds to non-interacting fermions without an impurity.

a superconductor. However, the energies do not provide a comprehensive understanding of particle-particle correlations, which is required to explain the physical mechanism behind the observed transition. Here, we provide further insight into the system by considering experimentally relevant probes that do not directly concern the energies[6]. These observables shed additional light on the details of the physics of our model.

In the following, we present the density profiles of the atomic clouds as well as density-density correlation functions (i.e., the shot noise) in real- and momentum space. These quantities may be measured, for example, by post-processing time-of-flight (TOF) images of few-body systems [48–51].

### 3.4.1   Spatial density profiles

An analysis of density profiles is an intuitive and common way of studying the behavior of trapped Fermi systems. For such an analysis, one calculates the spin-resolved density:

$$\rho_\sigma(x) = \sum_{k,l=1}^{n_b} \phi_k(x)\phi_l(x)\rho_{kl}^\sigma, \tag{14}$$

where $\rho_{kl}^\sigma = \langle\psi|k\rangle\langle l|\psi\rangle$ denotes the ground-state expectation values ($|\psi\rangle$ is the ground-state wave function) of the one-body density matrix, which encodes one-body correlations between single-particle orbitals, $\{\phi_k\}$, of the harmonic oscillator.

In Fig. 6, we show the density for the spin-up and spin-down particles as well as the total density for the $2\uparrow +2\downarrow$ systems with various impurity strengths $G$. For simplicity, the fermion-fermion interaction is omitted, i.e., $g=0$, since we observed that the features of the density profiles depend weakly on the considered values of $g$.

In Fig. 6 A, note a suppression of the $\uparrow$-density close to $x=0$ since the impurity repels particles of this spin type. In Fig. 6 B, we illustrate a peak in the $\downarrow$-density that grows with the value of $G$, indicating the presence of a bound state of a spin-down fermion with the impurity. Without a trap, at the single-particle level, such a bound state is described by an exponential

---

[6]It is worth noting that the use of the effective interaction in exact diagonalization leads to a fast convergence not only of the energy but also of other observables, in particular, of the one-body density matrix (see Ref. [31] for a benchmark). Therefore, we do not need to change the numerical routine in this section.

wave function

$$\psi(y_1) \sim \mathrm{e}^{-m|Gy_1|}, \tag{15}$$

which clearly exhibits a cusp at the origin. The corresponding density, $\rho(y_1) = |\psi(y_1)|^2 \sim$ $\mathrm{e}^{-2m|Gy_1|}$, inherits this cusp. It is impossible to have a cusp in the density in numerical calculations based upon the smooth basis set given by the harmonic oscillator eigenfunctions. Nevertheless, even in the restricted basis the emergence of the bound state becomes clearly visible in $\rho_\downarrow$. Note, that technically the suppression of $\rho_\uparrow$ around the origin should also exhibit a cusp, however, this is smeared out for the same reasons.

The emerging peak signals the local distortion of the wave function around the impurity and serves essentially as an intuitive illustration of the effect of the magnetic impurity amenable for experimental detection. However, to gain more insights into the physics that drives the transition presented in Fig. 5, one requires higher-order correlation functions that can show the effect of the interplay between $g$ and $G$. Below, we present the density-density correlators which provide insight into this interplay.

### 3.4.2 Density-density correlations

Here, we analyze density-density correlation functions in both real and momentum space. Pairing, which happens when the particles interact attractively, leaves distinct imprints on these quantities in momentum space, see for example Refs. [52–58] where relevant Fermi systems are discussed. Therefore, density-density correlation functions can further corroborate the physical picture of the few-body precursor of a QPT.

The density-density correlation function in real space is defined as

$$G^r_{\uparrow\downarrow}(x,y) = \langle \hat{\rho}_\uparrow(x)\hat{\rho}_\downarrow(y)\rangle - \langle \hat{\rho}_\uparrow(x)\rangle\langle \hat{\rho}_\downarrow(y)\rangle, \tag{16}$$

where $\hat{\rho}_\sigma(x) = \hat{\psi}^\dagger_\sigma(x)\hat{\psi}^\dagger(x)$ is the spatial density operator, whose expectation value we analyzed in the previous subsection. Analogously, in momentum space, we write

$$G^k_{\uparrow\downarrow}(p,q) = \langle \hat{n}_\uparrow(p)\hat{n}_\downarrow(q)\rangle - \langle \hat{n}_\uparrow(p)\rangle\langle \hat{n}_\downarrow(q)\rangle, \tag{17}$$

where $\hat{n}_\sigma(p) = \hat{\psi}^\dagger_{\sigma,k}\hat{\psi}_{\sigma,k}$ is the momentum-space density operator[7]. The quantities $G^r_{\uparrow\downarrow}$ and $G^k_{\uparrow\downarrow}$ vanish in the case of vanishing particle-particle interactions (statistically speaking the densities would be uncorrelated random variables). Therefore, the density-density correlation functions mark ideal candidates for investigating effects induced by interactions. We expect that correlations in momentum space are in particular useful for our purposes, since the BCS pairing is most easily visualized in momentum space.

In Fig. 7 A, we illustrate $G^r_{\uparrow\downarrow}$ for $g = -2.0$ for the $2\uparrow + 2\downarrow$ and $2\uparrow + 3\downarrow$ systems with $(G = 5.0)$ and without $(G = 0)$ the magnetic impurity. We see that the spatial correlations in $2\uparrow + 2\downarrow$ and $2\uparrow + 3\downarrow$ are similar. In the limit of strong impurity-fermion interactions $(G = 5.0)$, they both feature a suppression at the center of the trap. At the center there is no statistical correlation between the spin-up and the spin-down particles, and the system effectively partitions into two copies (left and right of the impurity respectively) with no cross-correlations between the sides, as anticipated from the discussion in 3.2.1.

---

[7]Note that $G^r_{\uparrow\downarrow}$ and $G^k_{\uparrow\downarrow}$ are not simply Fourier-transforms of each other. Fourier transform of the spatial density-density correlation function would give the structure factor.

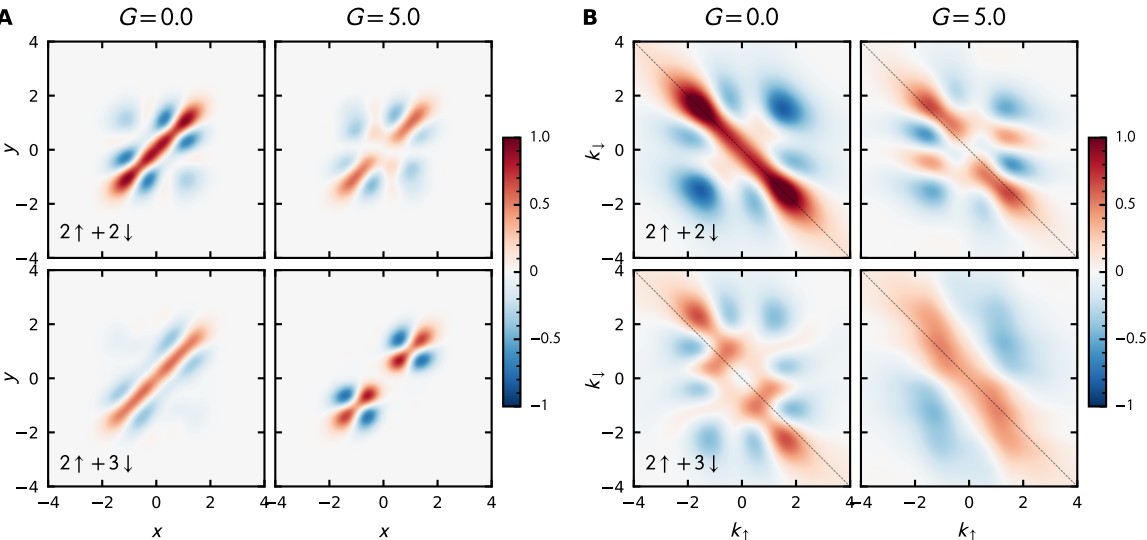

Figure 7: **Density-density correlation functions.** (A) Real space correlation function, $G_{\uparrow\downarrow}^r$. (B) Momentum-space correlation function, $G_{\uparrow\downarrow}^k$. All panels have $g = -2.0$, with ($G = 5.0$, right columns) and without ($G = 0$, left columns) the magnetic impurity. The top row is for the $2\uparrow + 2\downarrow$ system; the bottom row is for $2\uparrow + 3\downarrow$. The dashed lines in panel B show the anti-diagonal $k_\downarrow = -k_\uparrow$ where pairing in the balanced case is expected. In both panels red and blue values correspond to positive and negative correlation, respectively. The colormaps are in arbitrary units and normalized such that the uncorrelated values correspond to 0 (white) as well as maximal positive correlation corresponds to 1.

In Fig. 7 B, we illustrate $G_{\uparrow\downarrow}^k$ for the same system. The change of $G_{\uparrow\downarrow}^k(p,q)$ with $G$ highlights the cause of the few-body precursor of QPT in terms of pairing. For the $2\uparrow +2\downarrow$ system with $G = 0$, we see strong correlations on the anti-diagonal line $k_\downarrow = -k_\uparrow$ (indicated as a dashed line in Fig. 7 B). We interpret this as a precursor of the BCS mechanism, in which pairs are formed between the spin-up and spin-down fermions at equal but opposite momenta (i.e., at the zero center-of-mass momentum). For imbalanced systems, such as the $2\uparrow +3\downarrow$ system shown in the bottom left frame, pairing is expected to happen at the respective Fermi points (beyond the few-body regime) which results in the two maxima at $(\pm k_{F\uparrow}, \mp k_{F\downarrow})$ which are now offset with respect to the anti-diagonal.

In the presence of a strong impurity-fermion interaction (right column of panel B), the situation is opposite to that discussed in the previous paragraph. The balanced system exhibits two maxima that are slightly shifted away from the $k_\downarrow = -k_\uparrow$ line. In contrast, the $2\uparrow +3\downarrow$ system favors correlations along the anti-diagonal line. The reason for this behavior is the binding of a spin-down particle to the impurity, which turns the $2\uparrow +3\downarrow$ system into an effectively balanced system where BCS-like pairs can be formed.

# 4 Conclusions

In this section, we summarize our findings and give an outlook. We also provide a brief discussion of a possible experimental system.

## 4.1 Summary & Outlook

In this work we presented a numerical investigation of a few-fermion system in the presence of a spin-selective potential, which features some physics of a quantum phase transition driven by a magnetic impurity in the vicinity of an $s$-wave superconductor. In particular, one can lower the energy of the system by introducing a slight spin imbalance. This behavior is caused by the competition of pair formation due to the attractive fermion-fermion interaction and pair breaking due to the magnetic impurity. By tuning the strength of fermion-impurity interaction, one is able to study the crossover between different ground states.

Besides being of interest by itself, the system under study is a basic building block required to engineer more sophisticated set-ups including two or more impurities [59–62]. In such systems, for small enough spacing between the impurities, the bound states hybridize and form a sub-gap Shiba band [63, 64]. If in addition spin-orbit coupling is present in the host system such a setup may realize a topological superconductor. In this case, one can potentially observe Majorana edge modes at the ends of the hybridized impurity chains [65, 66]. Future works should focus on engineering cold-atom few-body experiments that can study the physics of one and two magnetic impurities, and complement studies of many-body systems, see, e.g., [67]. We outline some relevant ideas in the next subsection.

Finally, we note that in the present study, the number of particles was a fixed quantity precluding transitions between the sectors with $S = 0$ and $S = \frac{1}{2}$. To study quantum fluctuations typical for QPT's, one should consider a scenario in which these sectors are coupled. For example, this can be achieved if the spin-up and spin-down sectors are connected via a third hyperfine state that is not interacting. Time dynamics in this set-up may contain typical signatures of QPT's. Note that time-dependent simulations of few-body systems become possible [68, 69], allowing one to study time evolution in the vicinity of (few-body precursors

of) phase transitions, see, e.g., [70].

## 4.2 Experimental considerations

The Hamiltonian in Eq. (1) with $H_{\text{imp}} = 0$ is often used to model cold-atom experiments in quasi-one-dimensional geometries, see Ref. [6, 7, 71] and references therein. In particular, its accuracy has been tested for few-body systems of $^6$Li [17, 72]. The main new element of our work from the point of those experiments is the presence of a 'magnetic impurity', and we need to discuss it in some detail. The considered spin-selective potential may be natural for systems with large mass imbalance, e.g., $^6$Li-$^{133}$Cs, close to a favorable Feshbach resonance. This possibility was very recently discussed in Ref. [73]. Below, we shall briefly outline two other ideas for realizing $H_{\text{imp}}$.

One may engineer $H_{\text{imp}}$ by following the idea of an atomic quantum dot [74]. Here, the magnetic impurity is a single atom in a tight optical trap. For the idea to work, the impurity atom should be different from the spin-up and spin-down fermions. For instance, it could be a fermion in a different hyperfine state (not ↑ or ↓) or a particle with different mass (cf. [75]). In this implementation, the fermion-impurity interaction is of short range naturally, see Ref. [76] for more information. Furthermore, its strength can be tuned in a standard way using an external magnetic field. However, there are a few disadvantages of this approach: (i) Microtraps typically have $\mu$m-width, which implies that the zero-energy motion of the trapped atom needs to be considered. (ii) Deterministic preparation of such set-ups with a known (small) number of particles has not been demonstrated. The item (i) does not lead to a much more complicated analysis. By contrast, the item (ii) may lead to a typical many-body problem whose investigation we leave for future studies.

Alternatively, one may use spin-dependent potentials to mimic $H_{\text{imp}}$. For example, Ref. [77] demonstrates that it is possible to tune $g$ and $G$ independently of each other, and even realize $G_\uparrow = -G_\downarrow$ as we have in our calculations. The effects due to the finite width of spin-dependent potentials can be easily taken into account in our calculations. However, we expect that a finite width of the magnetic impurity does not change the main conclusions of this work, as long as this width is smaller than the length scale given by the Fermi momentum. Unfortunately, photon scattering induces losses, which are not included in the present theoretical model. Assuming that it will be possible to deterministically prepare a few-body system and a spin-selective potential, one could account for losses by post selecting time-of-flight images that have the desired number of atoms. In general, in the presence of losses, one needs to find a suitable experimental protocol. In particular, it seems natural to focus on $G_{\uparrow\downarrow}^k$, which should contain traces of pairing even in non-equilibrium. We leave a more elaborate investigation of this question to future studies.

## Acknowledgements

We acknowledge fruitful discussion with Areg Ghazaryan, Philipp Preiss, Selim Jochim and his group in Heidelberg. In addition, we thank Areg Ghazaryan for comments on the manuscript. We thank Pietro Massignan for sharing with us the data for benchmarking our numerical results without the impurity and Péter Jeszenski for support with implementing the TCM. Fig. 1 contains resources by *Pixel perfect* from Flaticon.com.

**Author contributions**   L.R. and D.H. contributed equally to this work.

**Funding information**   This work has been supported by European Union's Horizon 2020 research and innovation programme under the Marie Skłodowska-Curie Grant Agreement No. 754411 (A.G.V.); by the Deutsche Forschungsgemeinschaft through Project VO 2437/1-1 (Projektnummer 413495248) (A.G.V. and H.W.H.); by the Deutsche Forschungsgemeinschaft through Collaborative Research Center SFB 1245 (Projektnummer 279384907) and by the Bundesministerium für Bildung und Forschung under contract 05P21RDFNB (H.W.H). L.R. is supported by FP7/ERC Consolidator Grant QSIMCORR, No. 771891, and the Deutsche Forschungsgemeinschaft (DFG, German Research Foundation) under Germany's Excellence Strategy –EXC–2111–390814868. The work was partially supported by the Marsden Fund of New Zealand (Contract No. MAU 2007), from government funding managed by the Royal Society of New Zealand Te Apārangi (J.B.). We also acknowledge support by the New Zealand eScience Infrastructure (NeSI) high-performance computing facilities in the form of a merit project allocation.

# A    Extrapolation routine for the effective interaction

In this appendix, we briefly outline our routine for extrapolating the ground-state energies that we obtain with finite $n_b$ to the limit $n_b \to \infty$. To this end, we employ the method of least squares: we compute energies for several values of $n_b$ and fit them to the functional form

$$E(n_b) = E_\infty + \frac{a}{n_b^\sigma}, \tag{18}$$

where $a$ and $E_\infty$ are fit parameters and $\sigma$ is the exponent that determines the rate of convergence of the true ground-state energy with increasing the single-particle cutoff, $n_b$. In principle, when working with sufficiently many cutoff values, the best procedure is to include $\sigma$ as an open parameter. However, when the number of data points is rather small (as is the case for larger systems where values are expensive to obtain) this strategy will lead to extrapolated energies that are far from the exact solution due to only a few values in the scaling window. Often, a better choice here is to fix $\sigma$, ideally relying on some theoretical insights. As mentioned in the main text, for the case of bare interactions in 1D it was shown that, at leading order, ground-state energies converge to the infinite-basis limit with $\sigma = 0.5$ (see, e.g., [36]). The leading order exponent for the effective interaction is not known, however, empirically we observed that $\sigma > \frac{1}{2}$ and the best results have been obtained with $\sigma = 1.0$, as found by comparison to the transcorrelated method and the exact solution for the $1 \uparrow + 1 \downarrow$ problem. We therefore employ $\sigma = 1$ unless otherwise noted.

It is worth noting that not only the value of $\sigma$ but also the value of the prefactor $a$ is observed to be more favorable for the effective interaction, compared to the convergence properties for the bare interaction. In many cases, the favorable convergence properties would allow one to skip extrapolation $n_b \to \infty$ altogether within reasonable accuracy.

The results of our fitting procedure are summarized in Fig. 8: In panel A we show a comparison of the three extrapolation routines mentioned above for a value of $g = -4.0$, where it is apparent that the fixed value $\sigma = 1.0$ yields the best results. Panel B of the same figure shows the relative error of our extrapolated ground state energy for the $1 \uparrow + 1 \downarrow$ system

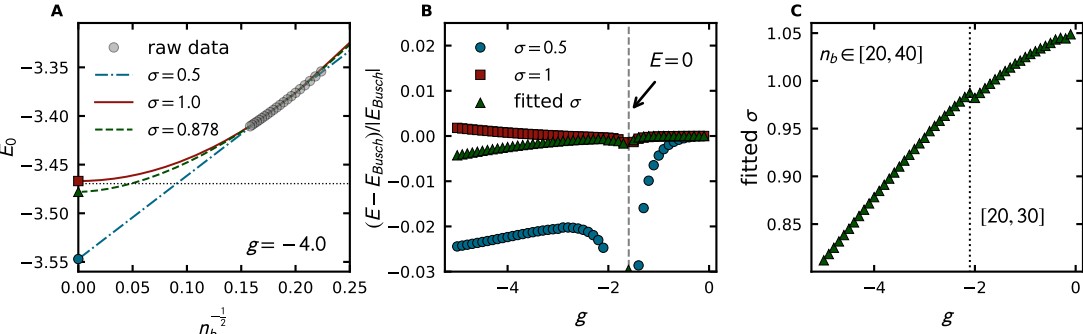

Figure 8: **Extrapolation of effective interaction results without magnetic impurity for the** $1\uparrow+1\downarrow$ **system**. (A) Extrapolation of the ground-state energy to the limit $n_b \to \infty$ for $g = -4.0$ for $\sigma = 0.5$ and $1.0$ (blue dashed-dotted and red solid curves, respectively) as well as for open $\sigma$ (green dashed curve). The black dashed line reflects the exact two-body solution. (B) Relative error of the extrapolated result with respect to the exact solution (color coding as in A). The vertical dashed line marks a zero-crossing of the ground-state energy. (C) Fitted values for $\sigma$ when it is unconstrained as a function of the interaction strength $g$.

with respect to the exact solution by Busch et al. [38] ($E_{Busch}$) as a function of the fermion-fermion interaction $g$. The large relative error at the gray line is caused by a zero-crossing of the energy. The low relative uncertainty further shows that $\sigma$ fixed to $1.0$ yields the best results for all considered couplings strengths.

Finally, in panel C of Fig. 8 the extracted values for an open $\sigma$ parameter are plotted for the same range of interaction strengths. For $g \gtrsim -2$, we used cutoff values in the range $[20, 30]$ for extrapolation. For stronger attractive interactions we used $[20, 40]$. Note that the parameter $\sigma$ in Fig. 8 C is close to 1.

## A.1 Extrapolation for systems with a magnetic impurity

In the presence of a magnetic impurity, i.e., when $G \neq 0$, slight complications arise. First of all, an odd-even staggering[8] as a function of the cutoff parameter $n_b$ is observed, which can be mitigated by separately extrapolating the values of odd and even cutoff values. The resulting extrapolated ground-state energies have been observed to lie within the achievable uncertainties.

Secondly, an interplay between the different interactions in the Hamiltonian may lead to a distinct behavior: for small impurity strength $G$ the data converges from above with increasing cutoff whereas at large impurity strength convergence from below is found. This behavior is shown for a $1\uparrow+1\downarrow$ system at $g = -2.0$ in panels A and C of Fig. 9. Moreover, this entails a region where results are virtually independent of the cutoff, as shown in panel $B$ of the same figure (note the tiny extent of the $y$-axis in all cases). In such a case the data may not be sufficiently fitted with a simple power-law and we therefore merely average the available datapoints to obtain our extrapolated result. This behavior is an artefact of the effective interaction approach, since the diagonalization with a bare interaction yields variational energies even with finite values of $G$ and hence should always display convergence from above.

---

[8]To understand this note that the odd basis functions cannot feel the potential at the origin.

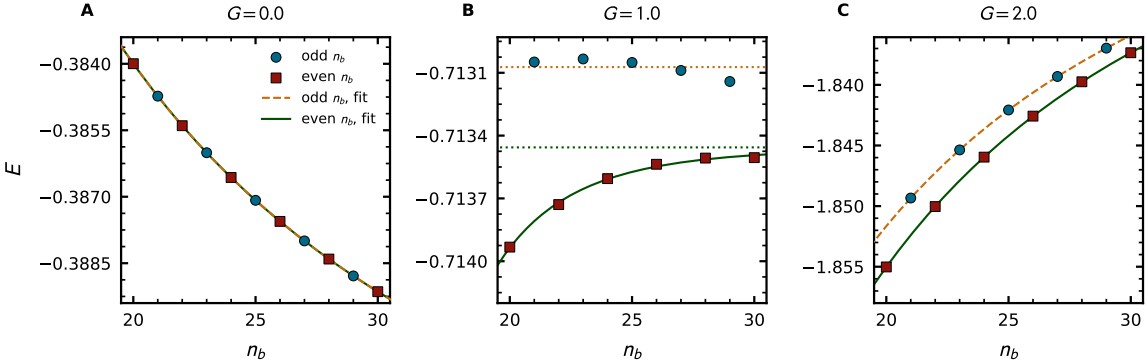

Figure 9: **Extrapolation of effective interaction results in the presence of a magnetic impurity for** $1 \uparrow +1 \downarrow$ at $g = -2.0$. (A) Convergence from above without the impurity. (B) Weak cutoff dependence in the vicinity of the crossover point, asymptotic values are indicated by dotted lines. (C) Convergence from below when $G$ is large.

## B    Additional data

In this appendix we present additional data for the few-body phase-diagrams at various cutoff values and particle numbers.

### B.1    Few-body phase diagram for individual cutoff values

To provide additional insight into finite-cutoff effects, we here compare the phase-diagram obtained with finite single-particle cutoffs to the extrapolated few-body (i.e., $1 \uparrow +1 \downarrow$ vs $1 \uparrow +2 \downarrow$) phase-diagram that was already shown in Fig. 3 of the main text.

As apparent from Fig. 10, the overall form of the phase-diagram does not change drastically at finite cutoff values (values between the lowest and highest cutoff $n_b$ are summarized by the green semi-transparent band) as compared to the extrapolated result. The latter is shown for extrapolations $n_b \to \infty$ with the coefficients $E_0 \propto n_b^{-0.5}$ (solid blue line) and $E_0 \propto n_b^{-1}$ (dashed red line), respectively. Although the exact values for the QPT differ slightly, in particular at intermediate particle coupling and impurity strength, the overall form is consistent between all versions.

In conclusion, within the effective interaction framework the qualitative features of the phase diagram do not depend strongly on the single-particle cutoff $n_b$ in the parameter range up to $|g| \sim 6.0$ at least. This observation carries over from the documented convergence properties of the effective interaction approach for more general few-body systems [31].

### B.2    Crossover between the $2 \uparrow +2 \downarrow$ and $2 \uparrow +3 \downarrow$ sectors

In addition to the analysis shown in the main text as well as in the preceding subsection, we here show some complimentary data for the few-body QPT in $2 \uparrow +2 \downarrow$ and $2 \uparrow +3 \downarrow$ sectors to highlight the similarities of this system to the smallest set-up with a single spin-up particle. In Fig. 11 we present $|\Delta_{32}|$, i.e. the energy difference between the lowest energy level of each sector, defined analogously to Sect. 3.2.2 of the main text. The limiting cases of $G > 0, g = 0$ and $G = 0, g < 0$ are shown in panels A and B, respectively. As already

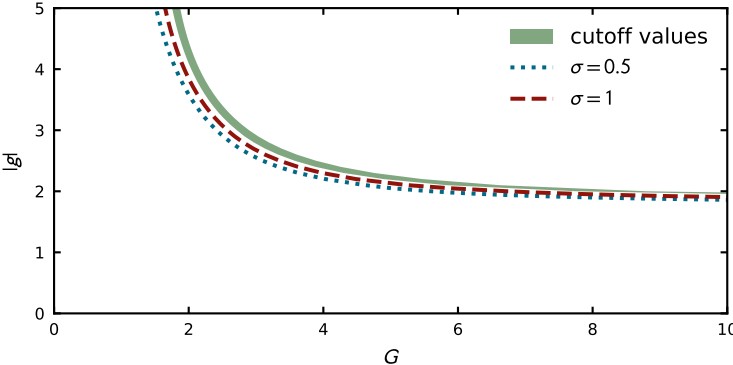

Figure 10: **Phase diagram for individual cutoff values.** Phase diagram for the $1\uparrow +1\downarrow$ vs $1\uparrow +2\downarrow$ for individual cutoff values shown by the shaded area. The upper edge of the area shows energies for cutoff 20, the lower edge for cutoff 30 ($g > -2.0$) and 40 ($g < -2.0$). The extrapolated results with the fixed exponent $\sigma = 0.5$ ($\sigma = 1$) is shown by the solid blue (dashed red) curve.

observed for the analogue system of fewer particles, there is no ground-state level crossing and the $2\uparrow +2\downarrow$ sector remains the lowest in the energy for all cases where one of the coupling strengths vanishes. In panel C of Fig. 11, the gap to the lowest energy level is shown for both sectors as a function of the impurity strength $G$ for constant particle interaction strength.

In addition, in panels B and C the different types of symbols reflect results at various single-particle cutoff values. The small spread in energy indicates that the effective interaction approach is well converged in the probed regime and produces quantitatively meaningful results. In the present case at $g = -3.0$, only at very large impurity strength one is able to distinguish the values for different cutoffs by eye on this scale.

Finally, in panel D of Fig. 11 we show the resulting few-body phase-diagram at a fixed cutoff of $n_b = 12$. The gray symbols reflect the actual data points, the arrows indicate the lines covered in panels A - C of the same figure.

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
