# Peer review of "Magnetic impurity in a one-dimensional few-fermion system"

_SciPost Physics_

## Round 1 · Referee Report · Anonymous (Referee 1) · 2022-5-21

Strengths
- Clearly formulated problem.
- A clear answer to the problem formulated.
- Careful numerical study.
- Results are a useful illustration of existing analytical theory.
- Results may be relevant to a cold-atom experiment.
Weaknesses
- Apparent lack of quantitative discussion and comparison of the obtained result with the bulk-limit analytical theory of Yu-Shiba-Rusinov
Report
This is a carefully constructed manuscript presenting the results of an exact-diagonalization study of a few-body system. The model, methods, and the results are clearly presented. The only improvement I may suggest is to attempt a comparison of the "phase transition" line in Fig. 4 with the phase transition line known from the Yu-Shiba-Rusinov model. It may require some work though to recast the parameters used in that model into the language (of g and G) utilized in the manuscript.
Requested changes
- Try to compare the results with those of the Yu-Shiba-Rusinov model -- please see the report for the details.

---

## Round 1 · Referee Report · Anonymous (Referee 2) · 2022-5-28

Report
This manuscript studies a few-body Hamiltonian for fermions interacting via a contact interaction in the presence of a magnetic impurity. The aim is to analyze the system as a precursor of a quantum phase transtion caused by the interplay between the interaction and impurity coupling strength. Such a theoretical study is relevant for ultracold atom systems, where the number of particles is well below the typically assumed number of particles in an actual many-body system.
The manuscript is very well written and the numerical results seem solid to me. It also has a clear relevance for the experimental community working on ultracold atoms. Despite all the positive aspects of this manuscript, I have as a reviewer to strictly adhere to the acceptance criteria of SciPost Physics. The closest match is criterion 4, "Provide a novel and synergetic link between different research areas", with criteria 1-3 not being fulfilled. I don't see criterion 4 being fulfilled either, but the authors could of course try to make their case better in the reply and revised version. Unfortunately, mere correctness and good quality writing is not enough to fulfill (at least one) of the four acceptance criteria of SciPost Physics. On the other hand, I would strongly recommend the publication of the current version of the manuscript in SciPost Physics Core.
The manuscript is very well written and the numerical results seem solid to me. It also has a clear relevance for the experimental community working on ultracold atoms. Despite all the positive aspects of this manuscript, I have as a reviewer to strictly adhere to the acceptance criteria of SciPost Physics. The closest match is criterion 4, "Provide a novel and synergetic link between different research areas", with criteria 1-3 not being fulfilled. I don't see criterion 4 being fulfilled either, but the authors could of course try to make their case better in the reply and revised version. Unfortunately, mere correctness and good quality writing is not enough to fulfill (at least one) of the four acceptance criteria of SciPost Physics. On the other hand, I would strongly recommend the publication of the current version of the manuscript in SciPost Physics Core.

---

## Round 1 · Referee Report · Anonymous (Referee 3) · 2022-5-30

Strengths
- well-structured
- the details of the calculations are well explained
- results are new
- experimental realisation is discussed
Weaknesses
- interpretation of the results
- too many trivial details in the main text
- does not satisfy the expectation criteria of SciPost Physics
Report
The manuscript discusses a few (2 to maximum 9) attractively interacting spinfull fermions trapped by an external potential. The external potential is superimposed on another narrow potential where a single fermion is trapped. The authors consider a zero-temperature behavior of the system depending on the interplay of two interactions: g - between the fermions themselves, and G - between the fermions and the so-called impurity in the narrow potential. The main result of their numerical (exact diagonalisation) calculation is the phase diagram which delineates between magnetic and nonmagnetic ground state. The authors find, for instance, a critical value of g=2 below which the system remains non-magnetic independent of the strength of G-interaction. This is an interesting result. The authors also consider density-density correlations to understand their findings better. All in all, I find the results valid and new, however, I do not quite agree with their interpretation. First of all, there is no resemblance to the BCS mechanism the authors mentioned. Also discussion of any Shiba-like states is not applicable for the case the authors consider. Also phase transitions arise as a result of ODLRO, which is far from being present in the system under consideration. So I am afraid I do not agree with the interpretation of the results and their direct relation to any phase transitions. However, on their own right the results are interesting and when shortened (e.g. parts about infinite interactions moved to Appendices, and discussion related to BCS superconductivity removed) the manuscript could be resubmitted to Phys Rev A or maybe SciPost Physics Core.

---

## Editorial Decision

resubmitted